# The Effect of Blood Contained in the Samples on the Metabolomic Profile of Mouse Brain Tissue: A Study by NMR Spectroscopy

**DOI:** 10.3390/molecules26113096

**Published:** 2021-05-22

**Authors:** Anastasia Glinskikh, Olga Snytnikova, Ekaterina Zelentsova, Maria Borisova, Yuri Tsentalovich, Andrey Akulov

**Affiliations:** 1The Federal Research Center Institute of Cytology and Genetics, Siberian Branch of the Russian Academy of Sciences, Academician Lavrentiev Avenue, 10, 630090 Novosibirsk, Russia; glinskikhav@gmail.com (A.G.); mariazolot@yandex.ru (M.B.); akulov.mri@gmail.com (A.A.); 2International Tomography Center, Siberian Branch of the Russian Academy of Sciences, Institutskaya str. 3a, 630090 Novosibirsk, Russia; zelentsova@tomo.nsc.ru (E.Z.); yura@tomo.nsc.ru (Y.T.); 3Faculty of Fundamental Medicine, Novosibirsk State University, Pirogova str. 2, 630090 Novosibirsk, Russia

**Keywords:** brain tissue, metabolomic profile, metabolomics, nuclear magnetic resonance

## Abstract

(1) Recently, metabolic profiling of the tissue in the native state or extracts of its metabolites has become increasingly important in the field of metabolomics. An important factor, in this case, is the presence of blood in a tissue sample, which can potentially lead to a change in the concentration of tissue metabolites and, as a result, distortion of experimental data and their interpretation. (2) In this paper, the metabolomic profiling based on NMR spectroscopy was performed to determine the effect of blood contained in the studied samples of brain tissue on their metabolomic profile. We used 13 male laboratory CD-1^®^ IGS mice for this study. The animals were divided into two groups. The first group of animals (*n* = 7) was subjected to the perfusion procedure, and the second group of animals (*n* = 6) was not perfused. The brain tissues of the animals were homogenized, and the metabolite fraction was extracted with a water/methanol/chloroform solution. Samples were studied by high-frequency ^1^H-NMR spectroscopy with subsequent statistical data analysis. The group comparison was performed with the use of the Student’s test. We identified 36 metabolites in the brain tissue with the use of NMR spectroscopy. (3) For the major set of studied metabolites, no significant differences were found in the brain tissue metabolite concentrations in the native state and after the blood removal procedure. (4) Thus, it was shown that the presence of blood does not have a significant effect on the metabolomic profile of the brain in animals without pathologies.

## 1. Introduction

In recent years, the number of applications of metabolomic tissue profiling for earlier diagnostics, monitoring the effectiveness of pathology therapy and assessments of the phenotype of living organisms has been rapidly increasing. One of the most powerful and versatile methods used in quantitative metabolomics studies is NMR spectroscopy [1,2]. The method of NMR spectroscopy is fast, non-invasive and provides highly reproducible results. Moreover, a conventional NMR experiment does not require a complex procedure for sample preparation. The analysis of chemical shifts and multiplicities of NMR peaks allows us to reference them to specific metabolites [3]. Thus, it is possible to identify and determine the concentrations of many metabolites in a single run [4,5,6,7,8]. NMR data provide great interest in the development of approaches to the metabolic phenotyping of living organisms [9,10]. Moreover, this method is suitable for identifying metabolites that can serve as markers of diseases such as cancer [11,12,13], diabetes [14,15,16,17], inborn metabolic disorders [18], and cardiovascular diseases [19,20].

An important step in metabolomic studies is the choice of suitable biological samples. Urine and serum/plasma are widely used biofluids for studies. [6,8,20,21,22,23,24,25,26,27,28,29] Their main advantages are the high range of detectable metabolites and their non-invasiveness or minimal invasiveness of sampling. However, metabolomic profiling of other tissues is becoming increasingly important in the field of metabolomics [5,7,30].

Despite a variety of significant advances in the field of NMR technology, there are still several issues related to factors that can influence the results. One of the features of metabolomic profiling of tissues is the presence of blood in the tissue samples. In the case of in vitro NMR studies, the presence of blood can distort the metabolomic profile of the tissue. In the tissue studies of laboratory animals such as rats and mice, a perfusion procedure allows us to remove blood from the sample. However, when the study requires many samples, this procedure is laborious and time-consuming.

Despite a large number of studies of tissues in the native state [31,32], there is no literature that considers how the presence of blood in the tissues affects their metabolomic profile. Thus, the purpose of this work was to evaluate the possible effect of blood contained in brain tissue samples on their metabolomic profile using the NMR spectroscopy method.

## 2. Results

The method of preparation of biological samples using a three-component mixture of water/methanol/chloroform allowed for the extract purifying from proteins and lipids and minimizing the baseline distortion in NMR spectra [4,33,34,35,36,37].

A typical NMR spectrum of the brain tissue of the CD-1 mouse line is shown in Figure 1. 36 metabolites were identified in the NMR spectra of the brain tissue. The concentrations of these compounds in the brain tissue were measured (Table 1). No significant differences in the concentrations between groups A and B were found by Student’s *t*-test for any of these metabolites. No multiple testing problem corrections, e.g., the Bonferroni correction, were applied.

## 3. Discussion

We reliably identified 36 metabolites and quantified the NMR spectra of extracts from the brain tissue of the CD-1 mice, which made it possible to make a comparison between the groups for the most abundant metabolites, which are often used for monitoring the pathological changes in biological and medical studies [21,23,33,34,35,36,37,38,39].

Deriving information on the metabolism of certain tissues may carry risks of sample contamination by other tissues and biological fluids of the organic body. Taking into account the structural and spatial organization of tissues may help us to avoid errors in the material intake, but separating all-penetrating biofluids from the tissues is often complicated. In this regard, a reasonable question arises about the potential contribution of biofluids to the errors in the analysis of the metabolism of tissue. In the metabolomic analysis by in vitro NMR, the vessels are involved in the volume of the sample taken, and can potentially lead to a significant error. In our work, we took the same sample volume (cerebral cortex tissue) from both groups of animals, so in the group of animals after the perfusion procedure, the entire volume of the sample contained only brain tissue, and in the group of animals in the native state, the sample consisted of brain tissue mixed with blood. The volume of blood in the brain of a mouse averages 3.5–4.0% of the total brain volume. However, the percentage of blood varies for different areas of the brain. For the cerebral cortex, this parameter is equal to 7.9% [40]. The result of our analysis did not reveal significant differences in the concentrations of the examined metabolites between groups of animals on the available sample size. This indicates that the concentrations of major metabolites in brain tissue and blood normally have a similar order of magnitude. One should also take into account that the percentage of blood in the brain tissue does not exceed 8%, so a small difference in the metabolomic compositions of brain and blood becomes negligible in the comparison of brain tissues with and without perfusion. However, it is still possible that certain low-abundant metabolites are present in blood, but absent (or almost absent) in the brain, and the levels of these compounds can vary under different pathophysiological processes. If that is the case, the presence of blood in the brain tissue may induce errors in the detailed metabolomic analysis. The question on the significant changes in the concentration of blood metabolites, which can distort the results of tissue metabolomic analysis, for example, in metabolic pathologies such as diabetes mellitus, requires additional research.

## 4. Materials and Methods

The laboratory animals used in the experiment were born, reared, and throughout the study were kept in the “Center for Genetic Resources of Laboratory Animals” SPF-vivarium of the Institute of Cytology and Genetics, Siberian Branch of the Russian Academy of Sciences (unique identifier RFMEFI62119X0023). We used 13 male laboratory CD-1^®^ IGS mice for this study. Initially, this line of mice was purchased from the Charles River Laboratories (Wilmington, MA, USA) in 2014. At the beginning of the experiment, the age of the animals was 8 weeks from the moment of birth. Mice were housed in open cages (OptiMice, Animal Care Systems Inc., Centennial, CO, USA) in groups of 3 or 4 individuals of the same sex throughout the study. The animals were provided with food and water consumption ad libitum with a standard autoclaved food ssniff^®^ R/MH autoclavable V1534-3 for rodents (Sniff Spezialdiäten GmbH, Soest, Germany) and purified sterile water with the addition of “Severyanka” mineral salts (Eco-Project LLC, St. Petersburg, Russia). Conditions for keeping laboratory mice are also included: illumination—14 light: 10 dark, temperature 22–24 °C, and relative humidity 40–50%.

We divided the animals into two groups to collect the biomaterial for spectroscopy. The first group of animals (*n* = 7) was subjected to the perfusion procedure (6 stages are described below). Group A: the second group of animals (*n* = 6), was not perfused. Group B: the sampling was limited to steps 1 and 6, but with the time for intake of material equal to group A.

The procedure for mouse perfusing and sampling consisted of the following steps: (1) the animal was anesthetized intraperitoneally by injection of a domitor (75 μL/10 g weight; Orion Pharma, Espoo, Finland) and subsequent injection of zoletil (60 μL/10 g weight; Virbac Sante Animale, Carros, France); (2) the animal’s chest was opened, exposing the heart; (3) a 5 mL syringe was filled with phosphate buffer pH 7.4 (KCl 2.7 mM, NaCl 140 mM, Phosphate 10 mM; AppliChem GmbH, Darmstadt, Germany) and attached to the catheter; (4) the needle of the catheter was inserted into the left ventricle of the mouse heart, the right atrium was cut with scissors; (5) 15 mL of PBS (approximately 10 times exceeding the volume of mouse blood) was injected for washing the circulatory system of the mouse. Perfusion was performed at the flow rate 5 mL/minute, so the procedure took 3 min. As was previously published, a required volume of PBS for transcardial perfusion of the mouse ranges from 10 mL up to 50 mL at a flow rate 5 mL/min or 7 mL/min respectively, which means that the whole perfusion procedure lasted from 2 to 7 min; [41]; (6) the mouse skull was opened, and the brain was removed. A fragment of the frontal cerebral cortex was separated and stored at −70 °C.

All experimental procedures were carried out in accordance with Directive 2010/63/EU of the European Parliament and the Council of the European Union of 22 September 2010 on the protection of animals used for scientific purposes, and approved by the Commission on Bioethics of the Institute of Cytology and Genetics SB RAS.

To obtain a protein-free extract of metabolites from the mouse brain, we used the following sample preparation protocol. Brain tissue was weighed and homogenized using a TissueRuptor II homogenizer (Qiagen, Venlo, The Netherlands) in a cold water/methanol/chloroform mixture in a ratio 1:2:2 (*v*/*v*; 1600 µL of solvent mixture per 150 mg of wet tissue), vortexed for 30 s, kept on ice for 10 min, and incubated at −20 °C for 20 min. The mixtures were centrifuged at 12,000 rpm, 4 °C for 30 min to pellet proteins. The top hydrophilic fraction was collected into fresh vials and lyophilized using a vacuum concentrator.

Dried extracts were re-dissolved in 600 µL of D_2_O (99.9%; Cambridge Isotope Laboratories Inc., Wilmington, MA, USA) containing 6 × 10^−^^6^ M DSS (sodium 4,4-dimethyl-4-silapentane-1-sulfonate; Cambridge Isotope Laboratories Inc., Tewksbury, MA, USA) as an internal standard and 20 mM deuterated phosphate buffer to maintain pH 7.4. The ^1^H-NMR measurements were carried out with the use of an NMR spectrometer AVANCE III HD 700 MHz (Bruker BioSpin, Germany) equipped with a 16.44 T Ascend cryomagnet. The proton NMR spectra for each sample were obtained with 96 accumulations. The temperature of the sample during the data acquisition was kept at 25 °C, the detection pulse was 90 degrees. The repetition time T_r_ between scans was 25 s to allow for the relaxation of all spins. The relaxation times T_1_ for some compounds are given in Appendix A and indicate that the condition Tr > 5T_1_ (for the majority of spins) or, at least Tr > 3T_1_ (for spins with longest relation times such as histidine) is fulfilled. Low power radiation at the water resonance frequency was applied prior to acquisition to presaturate the water signal. The pulse sequence zgpr was applied. The baseline correction and integration were done manually using the free demo version of the program MestReNova v12.0.

In a typical NMR spectrum of a blood serum extract, one can identify approximately 50–70 compounds (5, 6, 8, 33, 35); however, some of the signals are weak or partly overlapped by other signals; that can cause significant experimental errors in the integration of these signals. For this reason, only 36 metabolites giving strong and distinct signals in NMR spectra were chosen for the analysis. The attribution of signals which have been used for the metabolite quantification in the NMR spectrum is shown in Figure 1.

The signal assignment was performed using the human metabolome database [27] and our own experience in the metabolomic profiling of animal and human tissues and biofluids [21,23,33,35,36,37,38,41]. For some compounds, the signal assignment was confirmed by the addition of authentic compounds into samples (Appendix A). The concentrations of metabolites in the samples were determined by the peak area integration, with respect to the internal standard. Since the intensities of the NMR signals give information on the relative concentrations of molecules in the sample, for the determination of the absolute concentrations, an internal standard DSS with a concentration of 6 × 10^−^^6^ M was added to each sample. When identifying metabolites, the NMR signals were assigned to specific functional groups of molecules, which made it possible to determine the number of protons that contributed to each signal of the metabolite. The integral values were normalized to the number of protons corresponding to the integrated functional group. Since many metabolites have several NMR signals in different regions of the spectrum, the distinct signals with minimal overlapping by other signals were taken for the concentration determination (see Appendix A). The concentration of metabolites in tissue samples was calculated using the following formula:$$C_{M_{i}}\left(\frac{nmol}{g}\right)=\frac{I_{M_{i}}\times 9\times C_{DSS}\times V_{D2O}}{I_{DSS}\times n\times m_{tissue}},$$
where $C_{M_{i}}$ is the concentration of the metabolite, $I_{M_{i}}$ is the integral of the signal of the metabolite, 9 is the number of equivalent protons in DSS, $C_{DSS}$ = 6 × 10**^−^**^6^ M is the DSS concentration, ***V_D_*_2*O*_** = 600 μL is the volume of the NMR sample, $I_{DSS}$ is the integral of the DSS signal, ***n*** is the number of protons that contributed to the integrated signal of the metabolite, and ***m*_tissue_** is the sample mass (mg) of a tissue used for the sample preparation.

## 5. Conclusions

In the course of this work, we studied the brain tissue of CD-1 mice to assess the effect of blood on the metabolomic profile of a tissue sample in its native state. The method of high-field ^1^H NMR spectroscopy allowed us to reliably identify and quantify 36 different metabolites. Analysis of brain tissue metabolite concentrations in the native state and after the blood removal procedure showed that the presence of blood does not have a significant effect on the metabolomic profile of the brain in animals without pathologies. Based on the data obtained, it can be concluded that the metabolomic analysis of animal brain tissue does not require a laborious procedure for removing blood from the studied samples. At the same time, this is only valid for situations without significant changes in the blood metabolome; in the presence of such changes, additional studies are required.

## Figures and Tables

**Figure 1 molecules-26-03096-f001:**
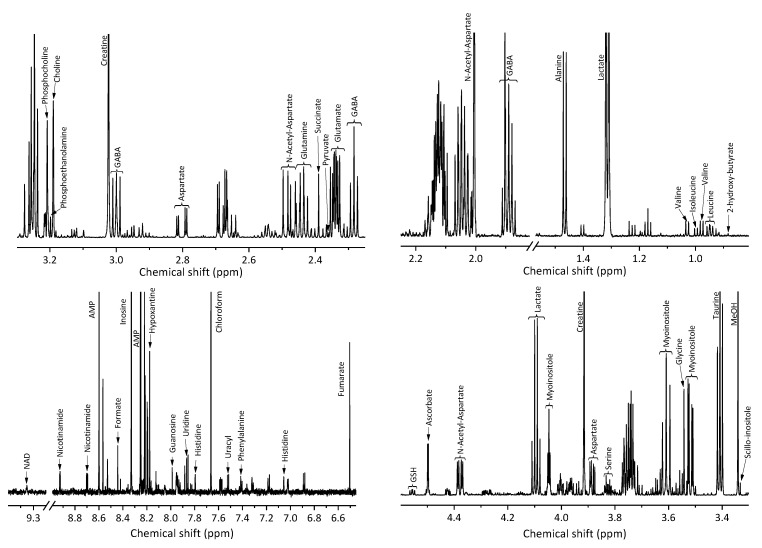
Typical NMR spectrum of brain tissue of the CD-1 mouse line with metabolite annotation.

**Table 1 molecules-26-03096-t001:** Metabolite concentrations (nmol/g) in the brain tissue of animals from groups A and B. The comparison of groups was performed by Student’s *t*-test with the estimation of the test power.

	Without Blood M ± S.D.	With Blood M ± S.D.	*p*	*Power*
2-hydroxy-butyrate	30 ± 25	30 ± 40	0.87	0.05
Alanine	650 ± 170	620 ± 160	0.75	0.06
AMP	510 ± 80	610 ± 140	0.15	0.32
Ascorbate	780 ± 140	790 ± 180	0.91	0.05
Aspartate	1590 ± 250	1810 ± 420	0.26	0.19
Choline	390 ± 120	340 ± 80	0.36	0.13
Creatine	4700 ± 1000	5100 ± 1000	0.57	0.10
Formate	80 ± 50	70 ± 30	0.75	0.07
Fumarate	70 ± 20	70 ± 20	0.92	0.05
GABA	1920 ± 540	1720 ± 530	0.52	0.09
Glutamate	5350 ± 840	6300 ± 1400	0.18	0.29
Glutamine	2190 ± 510	2370 ± 600	0.59	0.08
Glycine	690 ± 180	640 ± 150	0.65	0.08
GSH	530 ± 110	540 ± 130	0.89	0.05
Guanosine	30 ± 10	22.7 ± 9.7	0.46	0.23
Histidine	50 ± 10	50 ± 20	0.91	0.05
Hypoxanthine	160 ± 60	110 ± 30	0.10	0.44
Inosine	740 ± 280	560 ± 200	0.21	0.23
Isoleucine	30.4 ± 9.0	25.4 ± 7.2	0.30	0.18
Lactate	6300 ± 1300	6100 ± 1700	0.86	0.06
Leucine	50 ± 10	40 ± 10	0.09	0.38
*myo*-Inositol	3100 ± 650	3300 ± 740	0.63	0.08
*N*-Acetyl-Aspartate	3800 ± 640	4060 ± 810	0.56	0.09
NAD	22 ± 15	40 ± 20	0.09	0.39
Nicotinamide	90 ± 40	90 ± 40	0.97	0.05
Phenylalanine	40 ± 30	30 ± 10	0.38	0.13
Phosphocholine	300 ± 100	310 ± 80	0.91	0.05
Phosphoethanolamine	780 ± 190	820 ± 160	0.66	0.07
Pyruvate	21.2 ± 9.3	23 ± 10	0.80	0.06
scillo-Inositol	24.6 ± 6.6	27.1 ± 4.8	0.46	0.11
Serine	830 ± 240	750 ± 140	0.47	0.11
Succinate	200 ± 30	210 ± 90	0.68	0.06
Taurine	6100 ± 1300	6600 ± 1100	0.49	0.11
Uracyl	30 ± 15	19.8 ± 8.9	0.24	0.29
Uridine	90 ± 30	70 ± 25	0.15	0.22
Valine	60 ± 10	60 ± 10	0.87	0.05

List of abbreviations: AMP—Adenosine monophosphate; GABA—γ-Aminobutyrate; GSH—Glutathione reduced; NAD—Nicotinamide adenine dinucleotide.

## Data Availability

The metadata reported in this paper are available via https://www.ebi.ac.uk/metabolights/studyidentifierMTBLS2682 (accessed on 9 April 2022).

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
