# Peer review of "The Effect of Blood Contained in the Samples on the Metabolomic Profile of Mouse Brain Tissue: A Study by NMR Spectroscopy"

_molecules, 2021, doi:10.3390/molecules26113096_

Round 1

Reviewer 1 Report

The authors have addressed the raised issues. 

Author Response

Thank you for you review.

Reviewer 2 Report

The manuscript entitled: “The effect of blood contained in the samples on the metabolomics profile of mouse brain tissue: a study by NMR spectroscopy” reports an interesting study, by comparing the hydrophilic metatobolites of perfused brain tissue of mouse to the native one.

The design of experiment is well conducted, interesting and rigorous.

Aim of the work is a relative comparison (perfused versus native brain sample). However, the concentrations of metabolites reported in table 1, might be interpreted as absolute quantification but, to claim this, the inter-pulses delay (25s) should be ascertained to be at least 5 times the longest T1, for all the resonances used for quantification (table 1SI). The T1 of CH-OH of lactate is generally very long. Furthermore, the T1 changes with the pH and ionic strength of the solution, therefore it should be verified in the biological matrix.

Those aspects should be addressed before publication or, at least, better explained to improve the reader comprehension.

A minor revision is required before publication, according to the following list:

-Line 22 high- frequency 1H NMR? or high resolution 1H NMR?

-Lines 75-78 For a more reliable statistics also the power of the test for each metabolite should be included in table 1.

 -Table 1 SI and table 1 (Lines 75-78) The T1 of all the signals used for the quantification has to be measured in the biological matrix or a footnote should be added to table 1 to warn the reader for a possible underestimation in concentration of some metabolites.

 -Lines 72-73 The figure 1 reports only selected regions of the 1 H spectrum, reassembled in a picture. The whole 1H NMR spectrum has to be reported in the supporting information

-Line 157 2 in D2O should be subscript

 -Lines 157 and 233: alignment issues

-Line 159 1 in 1H should be superscript

-Line 164 please specify the power level of the 90 degree pulse and if it was optimized on each sample tube

 -Lines 165-166 please report the power level used for pre-saturation and if it was optimized on each sample tube

-Line 167-168 please specify the degree and type of function used for baseline corrections and, if the phase of the spectrum was performed manually or automatically. The type of integration, among those available with MestReNova, should be detailed (e.g. global spectra deconvolution, peak intensity, etc.).

Author Response

Thank you for your review. The response in attached file.

This manuscript is a resubmission of an earlier submission. The following is a list of the peer review reports and author responses from that submission.

Round 1

Reviewer 1 Report

The authors describe the influence of blood contamination on the metabolomic profile of brain cortex. The experiment design, experimental and analytical techniques are sound and indeed relevant to the field of metabolomics. Studies on the influence of blood remains within the tissues are indeed very rare and greatly needed. However, this manuscript would greatly benefit if other regions of brain could be included.

Reviewer 2 Report

The authors set out to determine if blood in brain tissue samples had an impact on the overall measured metabolic profiles of brain tissue extracts using NMR-based metabolic profiling analysis. The experimental design included 13 male mice, 7 of which received brain perfusion and 6 of which were not subjected to brain perfusion. Following perfusion, if carried out, the brain tissue was removed, weighed, homogenized, and metabolites extracted in a water/methanol/chloroform mixture. The hydrophilic layer was isolated, lyophilized and re-dissolved in D2O with a DSS standard and pH 7.4 phosphate buffer. The authors show excellent quality NMR data and were able to identify 36 metabolites commonly used for monitoring pathological changes in biological and medical studies. The authors reported concentrations of these 36 metabolites in nanomoles per gram of tissue +- a standard deviation. The authors reported p-values for each comparison and found no significant difference between the groups. The authors discuss the expected percent blood content of brain tissue to provide context for their observation that no difference was detected between perfusion and no perfusion groups.

Comments

Some details are not included in the manuscript that should be addressed. For example, the authors do not indicate whether or not any data normalization was applied prior to quantitation. The authors do not explain how quantitation was determined for metabolites that have many resonances. Did all of the resonances yield that same quantified values? Were all of the metabolites assigned with the same degree of confidence? The authors indicate that the assignments were made by addition of authentic compounds, reference to the human metabolome database, based on their prior experience. Do the authors include a table (supplementary) that specifies how each metabolite was assigned? Were all of the expected peaks observed for each metabolite? How many resonances were observed that were not able to be assigned? I.e. how complete were the spectral assignments? Have the authors submitted the raw data to a public data repository so that investigators can re-evaluate their results, including metabolite assignments and quantitation?

Reviewer 3 Report

This manuscript describes a study using mice which are sacrificed with and without removing all blood from the organism. Brain tissue is examined by metabolomics with and without blood. Bo differences are found.

This looks like a failed student project without any result and seems inappropriate to publish the lack of any signature. If you look at the righ level you have to find a difference for the presence of 10% blood.

I don't see any information value in this publication.

Reviewer 4 Report

The topic of the studies is important, but the presented data is not sufficient to draw the conclusions about the lack of influence of blood on the metabolome of brain tissue. The RSD is very high, from ca. 50% to way above 100%. The authors mentioned that blood volume is ca. 8% comparing to tissue volume, thus it is not possible to see the influence of blood components on brain tissue metabolome. There are many protocols and white papers recommending tissue perfusion before quenching and homogenization, including some which describe the effect of contamination with blood e.g. Analytica Chimica Acta 1153 (2021) 338300. 

There is lack of information about validation of the method used. The authors have to either give reference to the article, which includes validation of the protocol used in the current study or to provide validation results in Supplementary Materials. Another problem is the lack of quality control and normalization. These issues need to be properly addressed before considering the manuscript for publication.